# Room Temperature Detection of Hydrogen Peroxide Vapor by Fe_2_O_3_:ZnO Nanograins

**DOI:** 10.3390/nano13010120

**Published:** 2022-12-26

**Authors:** Mikayel Aleksanyan, Artak Sayunts, Gevorg Shahkhatuni, Zarine Simonyan, Hayk Kasparyan, Dušan Kopecký

**Affiliations:** 1Center of Semiconductor Devices and Nanotechnologies, Yerevan State University, Yerevan 0025, Armenia; 2Department of Computer and Control Engineering, Faculty of Chemical Engineering, University of Chemistry and Technology, 166 28 Prague, Czech Republic

**Keywords:** gas sensor, hydrogen peroxide vapor, iron oxide, magnetron sputtering, nanograins, zinc oxide

## Abstract

In this report, a Fe_2_O_3_:ZnO sputtering target and a nanograins-based sensor were developed for the room temperature (RT) detection of hydrogen peroxide vapor (HPV) using the solid-state reaction method and the radio frequency (RF) magnetron sputtering technique, respectively. The characterization of the synthesized sputtering target and the obtained nanostructured film was carried out by scanning electron microscopy (SEM), transmission electron microscopy (TEM), and energy-dispersive X-ray (EDX) analyses. The SEM and TEM images of the film revealed its homogeneous granular structure, with a grain size of 10–30 nm and an interplanar spacing of Fe_2_O_3_ and ZnO, respectively. EDX spectroscopy presented the real concentrations of Zn in the target material and in the film (21.2 wt.% and 19.4 wt.%, respectively), with a uniform distribution of O, Al, Zn, and Fe elements in the e-mapped images of the Fe_2_O_3_:ZnO film. The gas sensing behavior was investigated in the temperature range of 25–250 °C with regards to the 1.5–56 ppm HPV concentrations, with and without ultraviolet (UV) irradiation. The presence of UV light on the Fe_2_O_3_:ZnO surface at RT reduced a low detection limit from 3 ppm to 1.5 ppm, which corresponded to a response value of 12, with the sensor’s response and recovery times of 91 s and 482 s, respectively. The obtained promising results are attributed to the improved characteristics of the Fe_2_O_3_:ZnO composite material, which will enable its use in multifunctional sensor systems and medical diagnostic devices.

## 1. Introduction

Hydrogen peroxide (HP) is an oxygen–oxygen single bond compound primarily used as an oxidizing, bleaching, and antiseptic agent, and therefore, it is a significant substance used in pharmaceutical, biochemical, clinical, and food analyses. Trace detection of low HPV concentrations is a current issue in the development of non-invasive diagnostic systems for precisely monitoring human exhaled air at the ppm to ppb level. The quantitative analysis of HPV-containing environments can be carried out by numerous techniques, such as chromatography, electrochemistry, fluorescence, chemiluminescence, chemosensory, etc. Despite the variety of methods, resistive sensors designed based on chemosensory principles (so-called chemoresistors) are advantageous due to their high sensitivity and stability, low power consumption, and relatively easy fabrication technology [1,2,3,4,5]. 

Metal oxide semiconductors (MOSs) are advanced multifunctional materials considered as the main candidates for use in resistive gas sensors due to their high sensitivity, chemical and temporal stability, wide and tunable band gap, simple fabrication technology, and cost-effectiveness [6,7]. One of the most widely used MOSs in gas sensors is iron oxide, which is an inorganic compound with the chemical formula of Fe_2_O_3_. It appears in two polymorphic forms, *α* and *γ*, of which α-Fe_2_O_3_ (hematite) is an inexpensive, common, eco-friendly, and non-toxic material with a tunable band gap (∼2.1 eV). The most stable form of iron oxide, hematite, exhibits excellent gas sensing characteristics, and hence, it is considered one of the main candidates for use in this field. Furthermore, electrical, chemical, and gas sensing properties of α-Fe_2_O_3_ can be notably improved by the functionalization of the material using dopants (or solid solutions) in the bulk and noble metal nanoparticles as a surface enhancement [8,9,10,11]. 

In the last decade, ZnO-based nanostructures have become one of the most widely used metal oxides in the manufacturing of nanostructured gas sensors because of their unique sensing behavior, high electron mobility (~400 cm^2^ V^−1^ s^−1^), wide bandgap energy (3.37 eV), and large excitation binding energy (60 meV). Moreover, nanocomposite structures based on multicomponent materials show much better gas-sensing results than the pure materials themselves [12,13,14]. For instance, iron oxide nanorods were synthesized and decorated with zinc oxide nanoparticles using the hydrothermal method [15]. A response enhancement of approximately 59% was registered toward toxic gases/alcoholic vapors for ZnO-decorated α-Fe_2_O_3_ nanorods, along with a faster response rate compared to that of the pure α-Fe_2_O_3_ nanorods. Fan et al. [16] synthesized a Fe_2_O_3_/ZnO heterostructure using the atomic layer deposition (ALD) technique. The Fe_2_O_3_/ZnO heterostructure showed a better response (133.1) to 100 ppm H_2_S compared to the Fe_2_O_3_ nanosheet response (8.7) at 250 °C. Li et al. [17] synthesized Au/Fe_2_O_3_-ZnO gas-sensing materials by combining co-precipitation and microwave irradiation processes. The Au/Fe_2_O_3_-ZnO based sensor showed a high sensitivity (154) to acetone vapor at 270 °C, while the selectivity could be tuned by adding the Fe_2_O_3_ content to the ternary materials.

Different techniques have been applied to obtain nanostructured thin films for gas sensor application, such as sol-gel, chemical bath deposition (CBD), pulse laser deposition (PLD), polymer-assisted deposition, RF magnetron sputtering, etc., among which the RF sputtering method provides a good film uniformity and a high deposition rate at low temperatures [18,19,20,21].

Herein, we designed and characterized a Fe_2_O_3_/ZnO nanostructured sensor obtained by the RF magnetron sputtering method. We also studied the gas sensing behavior toward HPV under UV irradiation at temperatures ranging from 25 to 250 °C. The measured sensing results demonstrated high sensitivity and selectivity to low concentrations of HPV at RT.

## 2. Experimental

### 2.1. Gas Sensor Fabrication

First, the Fe_2_O_3_:ZnO ceramic cylindrical target was prepared by the solid-state reaction method for magnetron sputtering. The technological steps of this procedure were presented in more detail in our previous works [22,23]. For this purpose, α-Fe_2_O_3_ and ZnO (wurtzite) nanopowders (nanopowders, 20–40 nm, Alfa Aesar, Haverhill, MA, USA) with a purity of 99.9% were used to synthesize the cylindrical Fe_2_O_3_:ZnO target, with a thickness of 2 mm and a diameter of 50 mm, containing 20 wt.% of ZnO (Figure 1a). 

Active films for the detection of HPV were obtained by the RF magnetron sputtering method using the VTC-600-2HD DC/RF Dual-Head High Vacuum Magnetron Plasma System (Figure 1a). Magnetron sputtering allows for the deposition of a very wide range of materials, including metals, dielectrics, ceramics, etc., as an advanced vacuum coating technique for obtaining nanostructured thin films [24]. Here, we used the synthesized Fe_2_O_3_:ZnO ceramic target after mechanical and chemical treatment. The main mechanism of magnetron sputtering methods is shown in Figure 1b. The sputtering target is bombarded with energetic ions of inert gases that are present in the pre-ignited plasma environment. The dynamic striking of these energetic ions on the water-cooled target emits nanograins of the target material condensing on the substrate as a deposited nanostructured layer. We used a high purity (99.99%) argon (Ar) as the most preferred ‘inert’ or ‘noble’ gas, considered non-explosive when subjected to the RF (13.56 MHz) magnetic field.

At the beginning of the sensor fabrication, we ordered factory-designed Multi-Sensor-Platforms (TESLA BLATNÁ, Blatná, Czech Republic) as sensor substrates (Figure 2). This sensor platform contains interdigital electrode structures (IDES), a heater, and a temperature sensor (Pt 1000) on an alumina substrate [22]. The Fe_2_O_3_:ZnO sensing layer was deposited onto the IDES, converting the platform into an HPV sensor. Then, the surface of the Fe_2_O_3_:ZnO based film was sensitized with the palladium catalytic nanoparticles deposited by DC sputtering method. Generally, these nanoparticles create spillover zones around themselves on the surface of the gas sensing layer, where the mechanism of chemical and electrical sensitization takes place. Target gas molecules arriving in such zones are more easily dissociated, which leads to improved sensitivity and response times. The technological regimes of the sputtering processes are summarized in Table 1. In the final step of sensor manufacturing, the sensor platform with the Fe_2_O_3_:ZnO film and the Pd nanoparticles was annealed for 3 hours at 350 °C to stabilize its performance parameters.

### 2.2. Gas Sensing Test

The HPV sensing behavior was investigated using a self-designed gas testing setup including a 2 L gas chamber, a Keithley DMM7510 7 1/2 Digital Multimeter, a Keithley 2231A-30-3 DC Power Supply, a UV LED (*λ* = 365 nm, with the illumination of 3 mW cm^−2^), a heater, and an air circulator fan (Figure 3). To obtain the HPV concentration in the gas chamber, a desired amount of the liquid hydrogen peroxide was dropped on the purpose-built evaporating crucible, converting liquid hydrogen peroxide to HPV. The real concentrations of HPV were calculated by considering the percentage of liquid hydrogen peroxide aqueous solution, the amount of dripped drops, and the chamber volume [25]. For the measurements of resistance variation over time, a 3 V DC voltage bias was applied to the IDES, while a bias voltage range of 0–5 V was used for the sensor heater to obtain operating temperatures ranging from 25 to 250 °C.

HPV leads to an increase in the sensor resistance for n-type semiconductors (Fe_2_O_3_:ZnO); therefore, the gas response was defined as yielding values greater than one: *S* = *R_gas_/R_air_*, where *R_gas_* and *R_air_* are the sensor resistances in the HPV environment and in the air, respectively.

### 2.3. Characterization

For magnetron sputtering processes, the morphology of the target material is quite important because the bombardment of energetic ions on the target surface, with different porosity and granularity, leads to different sputtering rates. Therefore, the morphological structure of the Fe_2_O_3_:ZnO target was investigated by scanning electron microscopy using the MIRA 3 LMH (Tescan) instrument under 15 kV of accelerating voltage (Figure 4). It is evident from the SEM images that the average pore sizes in the target material are in the range of 5–10 µm, which is large enough that energetic ions can easily diffuse in the pores, slowing down the continuing sputtering process. Despite this, one of the advantages of the high-frequency sputtering process is that the energetic ions attach and detach from the surface of the target material at a high frequency (13.56 MHz) and do not manage to diffuse into the pores. Moreover, the high-frequency electromagnetic field mainly prevents the accumulation of positive ions on the target surface, ensuring the continuity of the sputtering process [26].

Generally, magnetron-sputtered films have a granular structure, and the size of the grains in these films is largely determined by the sputtering conditions [27]. SEM images of the sputtered Fe_2_O_3_:ZnO film were also obtained to confirm the granular structure of the film (Figure 5). The granular network is quite homogeneous, ranging from 10 to 30 nm in diameter, except for the presence of some cracks with a size of about 10–20 nm. It is assumed that these cracks are due to the film annealing as a result of the difference between the thermal expansion coefficients of the film and the sensor substrate, or as a result of so-called radiation damage, which is created by the electron microscope beam on the fine surface of the films at a higher magnification, and this cannot be easily avoided. In any event, these cracks did not affect the temporal stability of the sensor in a practical manner because the gas sensing parameters (mainly the sensor response) were tested over a long-term interval, and no significant drift was recorded.

The Fe_2_O_3_:ZnO target material and the sputtered film were also well studied by the EDX elementary analysis. The availability of Fe, Zn, and O peaks both in the target material and in the film obviously present the weight and atomic percentage of these elements (Figure 6). The real concentrations of Zn in the target material and in the film are 21.2 wt.% and 19.4 wt.%, respectively, faintly deviating from the initial calculated value (20 wt.%). Such deviations are assumed to be caused by the errors of the measuring instruments and the peculiarities of the technological processes. 

EDX spectroscopy yielded the e-mapped images of the Fe_2_O_3_:ZnO film confirming the uniform distribution of O, Al, Zn, and Fe elements, with the distinctive characteristic lines of X-ray intensity (Figure 7).

The crystal structure of the Fe_2_O_3_:ZnO material was analyzed by TEM, and the results are depicted in Figure 8. The hexagonal shape of ZnO [28], with average particle sizes of 15–30 nm in diameter, was confirmed (Figure 8a,b). The lattice fringes of the Fe_2_O_3_ and ZnO materials are clearly visible in the high-resolution TEM (EFTEM Jeol 2200 FS, JEOL Ltd., Tokyo, Japan) image of the Fe_2_O_3_:ZnO material (Figure 8c), with the interplanar spacing of 0.5 nm and 0.25 nm, respectively [29]. The selected area electron diffraction (SAED) pattern of the material revealed the nearly polycrystalline structure of the Fe_2_O_3_:ZnO material [30].

## 3. Results and Discussions

### 3.1. Gas Sensing Properties 

The gas sensing characteristics of the Fe_2_O_3_:ZnO sensor at the different operating temperatures to detect the various concentrations of HPV, with and without UV irradiation, were thoroughly investigated. In the initial stage of the research, the gas sensing parameters under dark conditions were examined in temperatures ranging from 25 to 250 °C at the HPV concentrations of 3–225 ppm, confirming its extremely high sensitivity, even without UV assistance. The temperature dependence curve of the sensor response appeared with a pronounced maximum peak at 150 °C, corresponding to a response value of 2600 (Figure 9a). The sensor satisfactorily responded to HPV (*S* = 23), even at RT, proving the existence of the chemisorption phenomena at this temperature, without additional thermal support. The dynamic resistance curves of the sensor for the seven different concentrations of HPV were measured at 150 °C, representing a rather low detection limit of HPV (3 ppm), with a response value of 300 (Figure 9b). Distinct response/recovery curves were plotted at each successive concentration measurement. As the HPV concentration increased up to 120 ppm, an almost linear characteristic appeared, while at higher concentrations, a near-saturation trend was observed (Figure 9c). For the high performance and practical applicability of the sensing device, it is extremely important to evaluate the response and recovery times as a function of the operating temperature. Thus, the dependence of the response and recovery times on the operating temperature was obtained (Figure 9d). Despite the sensor’s reasonable response at RT, the response and recovery times at this temperature were tens of minutes, demonstrating a fairly inefficient sensor performance, while at higher temperatures (>200 °C), they reached the order of seconds. This behavior is predictable because as the temperature increases, the rates of chemical reactions and gas diffusion tend to increase steadily [31]. 

One of the most common and effective ways to reduce the response/recovery times and operating temperature of the sensor is revealed by exposing its active surface to UV light [32], thus the gas sensing characteristics of the sensor in this study were tested and investigated under UV irradiation. First, the change in the real-time resistance of the sensor was observed when illuminated with UV light. As shown in Figure 10a, the sensor resistance in the presence of UV irradiation dropped more than ten times, reducing from 63 kΩ to 600 Ω, while exhibiting an extremely short UV response time (~8 s). By continuing to keep the sensor under UV rays, the sensor baseline resistance successfully stabilized, and all subsequent investigations were performed under these conditions.

The responses of the Fe_2_O_3_:ZnO sensor were recorded at different temperatures (25–250 °C), combined with UV irradiation toward 1.5 ppm HPV (Figure 10b). Starting from RT, the sensor response decreased persistently, becoming somewhat stable from 200 °C, which definitely prompted us to choose the RT as the operating temperature. It is assumed that the effect of UV rays creates favorable conditions for gas adsorption and further interaction with the lattice [33], and the use of UV rays, combined with temperature heating, leads to the acceleration of the HPV desorption rate, producing a significant decrease in the response. Moreover, without thermal excitation, the presence of UV light on the film surface is sufficient to create light-generated free electrons in the conduction bands of both Fe_2_O_3_ and ZnO materials, which will also lead to a decrease in the barriers between the presumably formed nanojunctions. This will allow for the tuning of the resistance of the Fe_2_O_3_:ZnO nanocomposite material, bringing it closer to the measurable range. Thus, the sensor showed almost the same response at RT, with and without UV irradiation, to 3 and 225 ppm HPV, respectively (Figure 10c). The presence of UV light on the sensor surface resulted in not only a significant improvement in the sensor response, but also an increase in the response and recovery rates. It is assumed that UV-generated electron/hole pairs intensively participate in the chemisorption processes on the semiconductor surface and in parallel, UV rays also stimulate the rate of electron exchange in chemical reactions. The sensor’s real-time repeatability—as one of the most important determinants of temporal stability—at every successive response measurement demonstrates its suitability for use in real environments. Therefore, at the same 1.5 ppm HPV concentration, the dynamic responses of the sensor were recorded by performing six different measurements, and the results are presented in Figure 10d. The sensor consistently showed a stable response, with nearly identical response/recovery curves (the relative deviation of response values: 0.97–1.75%) obviously confirming the sensor’s high repeatability behavior. 

The real-time resistance curves of the Fe_2_O_3_:ZnO sensor in the range of 1.5 to 56 ppm HPV were also demonstrated under the UV light, corresponding to the response values of 12 to 1930, respectively (Figure 11a). It is obvious that the presence of UV rays on the sensor surface not only improves the sensitivity and speed, but also reduces the low detection limit from 3 ppm to 1.5 ppm, clearly justifying their use as an alternative to traditional heating. The linearity of the sensor is a very important characteristic for the accurate estimation of the target gas concentration, as well as for its easy calibration processes. The dependence of the sensor response on the HPV concentration is approximately linear (Figure 11b), making it a promising detector in the HPV concentration range of 1.5 ppm to 56 ppm; thus, it can be successfully used, for example, in non-invasive diagnostic systems for respiratory diseases. The recovery and response times, considered typical indicators of the sensor’s performance, are significantly improved in the presence of UV irradiation combined with thermal heating. As shown in Figure 11c, the registered response and recovery times decrease with an increase in sensor temperature. Even at RT, the sensor’s response and recovery times were within the acceptable ranges of 91 s and 482 s, respectively (Figure 11d). The relatively high recovery time is assumed to be due to the rather difficult desorption processes of HPV molecules from the porous (granular) film, which leads to difficulty in reaching the sensor’s baseline resistance level.

The sensor selectivity was evaluated by testing its response to six different gases, including 130 ppm ammonia, 400 ppm acetone, 650 ppm ethanol, 3200 ppm water vapor, 320 ppm toluene, and 350 ppm DMF (dimethylformamide), compared to a 1.5 ppm concentration of HPV (Figure 12a). Despite the fairly high concentrations of reference gases (130–3200 ppm), the sensor showed a relatively higher response to ammonia vapors (Figure 12b), which is about 4.9 times lower than that of HPV. This is definitely a sufficient result to consider the sensor as a device capable of working in real environments, as it is endowed with high selectivity. 

The gas sensing results of our Fe_2_O_3_:ZnO based sensor were compared with the characteristics of currently available researched sensors to confirm the relevance and scientific novelty of the obtained results. The comparison table includes the type of sensing materials, HPV concentrations, operating temperatures, and response values, which invariably play a decisive role in sensor performance (Table 2). Our sensor showed clearly comparable gas sensing characteristics, in which rather high response values were most noticeable.

### 3.2. HPV Sensing Mechanism

It is known that the gas sensing mechanism of resistive type sensors is mainly determined by surface kinetic processes, which include gas adsorption/desorption processes, diffusion of molecules on the surface and into the porous film, as well as chemical reactions. The change in film resistance in the presence of a target gas is mainly due to the surface reactions, as a result of the particular concentration of the localized oxygen ions in the surface changes. Moreover, the surface and bulk diffusion of the oxygen species are greatly facilitated by the oxygen vacancies on the metal oxide surface. Initially, under normal atmospheric conditions, neutral oxygen species (O, O_2_) are physiosorbed on the semiconductor surface, after which chemisorption is observed, accompanied by their transformation into localized ions due to free electrons taken from the conductance band of the semiconductor. The supposed chemical reactions characterizing these processes are presented in Figure 13a. Under these conditions, an electron depletion layer is formed on the surface of the Fe_2_O_3_:ZnO nanograins, and the corresponding electrical resistance is established. Here, double Schottky barriers are formed at the intergranular junction of the Fe_2_O_3_:ZnO nanograins [22,43,44]. 

When HPV molecules are adsorbed on the semiconductor surface, they dissociate into water and oxygen molecules. Water molecules are usually easily desorbed from the surface, while oxygen molecules chemically interact with the semiconductor surface, taking even more free electrons from its conduction band, leading to an increase in the number of localized oxygen ions on the semiconductor surface (the corresponding supposed chemical reactions are shown in Figure 13b). This is reflected in an increase in the double Schottky barriers, leading to an increase in the semiconductor resistance. As a result, such a change in film resistance is typically referred to as a sensor response.

At high temperatures (>200 °C), O^−^ and O^2−^ oxygen species predominate on the sensor surface, whose further participation in the kinetic phenomena leads to a decrease in the sensor response in the presence of UV rays (Figure 10b). At RT (<200 °C), O_2_^−^ oxygen species play the main role in the gas sensing mechanism of the Fe_2_O_3_:ZnO sensor, leading to rather high sensitivity. This was observed because the UV rays not only induced the physisorption of HPV molecules on the Fe_2_O_3_:ZnO surface, but also stimulated the chemisorption processes [43]. 

The excellent combination of Fe_2_O_3_ and ZnO materials results in a higher photocatalytic activity of the Fe_2_O_3_:ZnO composite compared to that of the pristine Fe_2_O_3_ and ZnO materials, which is reflected in the high performance of the Fe_2_O_3_:ZnO sensor under UV irradiation. One of the reasons for this improvement may be that the conduction band level of α-Fe_2_O_3_ is lower than that of ZnO, which leads to the transformation of ferric ions in α-Fe_2_O_3_ to ferrous ions by the capturing of electrons. Due to the light activation, the excited electrons are transferred from the conduction band of ZnO to that of α-Fe_2_O_3_. In this case, the α-Fe_2_O_3_ will serve as a unique sieve for photogenerated electrons, which will lead to the spatial separation of charge carriers and a decrease in the probability of their recombination [45].

The illumination of UV-assisted resistive sensors is the best way to activate chemical reactions on the MOS surface and reduce the sensor baseline resistance, making this process an excellent alternative to energy-demanding heating. It is assumed that UV irradiation actively affects the adsorption/desorption processes of HPV on the Fe_2_O_3_:ZnO surface, allowing chemical reactions to occur without additional thermal stimulation. In these conditions, more neutral oxygen species absorbed on the surface become ions, resulting from the presence of light-generated electron/hole pairs. These ions greatly contribute to gas sensing processes by steadily improving gas sensitivity, selectivity, and response/recovery times [46,47,48].

## 4. Conclusions

In this study, we developed an RT HPV sensor based on the Fe_2_O_3_:ZnO nanocomposite material. The characteristics of the Fe_2_O_3_:ZnO material were revealed, and the gas sensitivity parameters were thoroughly investigated. The interplanar spacing of Fe_2_O_3_ and ZnO materials was 0.5 nm and 0.25 nm, respectively, and the SAED pattern confirmed the nearly polycrystalline structure of the Fe_2_O_3_:ZnO material. The maximal response was registered at 150 °C, atm which the Fe_2_O_3_:ZnO sensor displayed actual high response values of 42 and 2600, corresponding to 3 and 225 ppm HPV, respectively, even without UV assistance. The sensor responded quickly enough (~8 s) to UV illumination and under these conditions, the operating temperature of the sensor was reduced to RT, and the low detection limit was lowered to 1.5 ppm, while reducing the response/recovery times. The sensor showed linear response dependence on the HPV concentration in the range of 1.5 ppm to 56 ppm. The sensor selectivity was tested in the presence of different environmental gases, and rather a high selectivity behavior was confirmed. The design and fabrication of an HPV sensor with high sensitivity, repeatability, and selectivity to other interfering gases can be achieved through the Fe_2_O_3_:ZnO nanograins-based composite material, which makes it a perfect candidate for low-concentration HPV sensor application.

## Figures and Tables

**Figure 1 nanomaterials-13-00120-f001:**
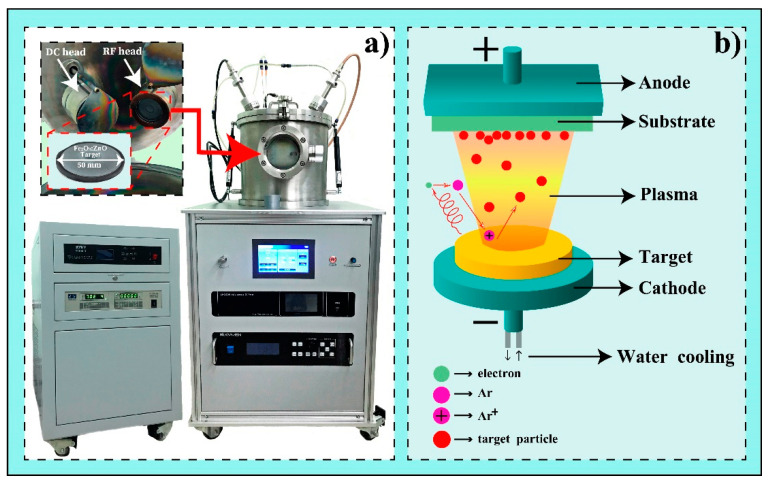
(**a**) Actual photographs of the Fe_2_O_3_:ZnO sputtering target and the VTC-600-2HD DC/RF Dual-Head High Vacuum Magnetron Plasma System; (**b**) representation of the magnetron sputtering operating mode.

**Figure 2 nanomaterials-13-00120-f002:**
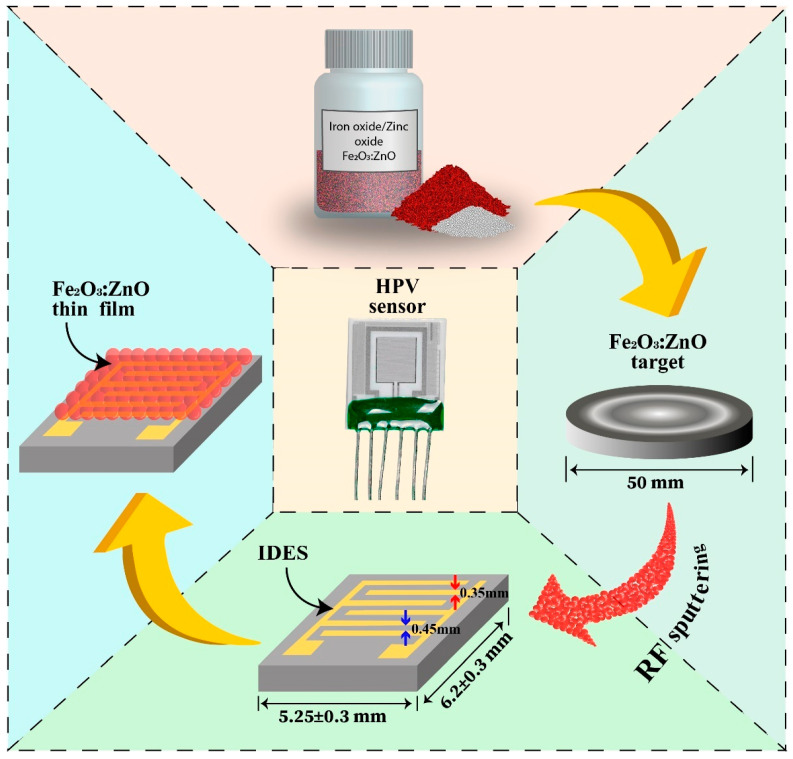
Schematic block diagram of the HPV sensor fabrication.

**Figure 3 nanomaterials-13-00120-f003:**
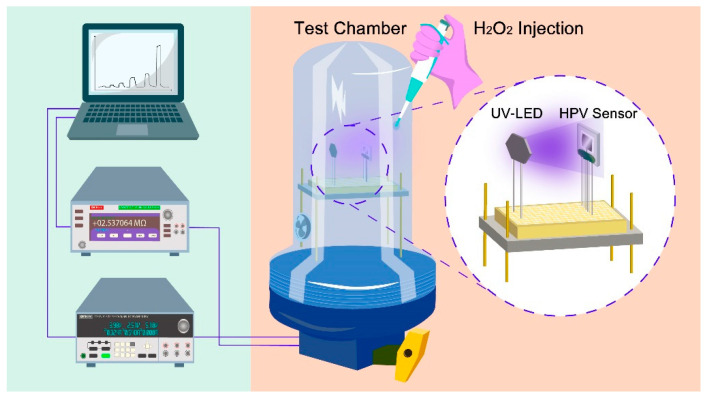
Schematic block diagram of the HPV testing system.

**Figure 4 nanomaterials-13-00120-f004:**
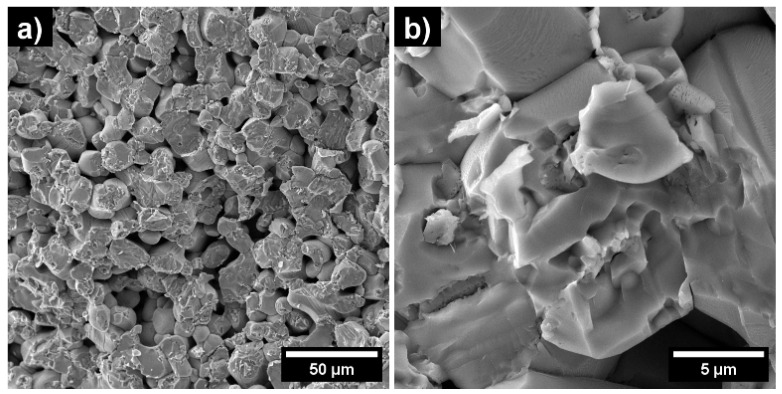
SEM images of the Fe_2_O_3_:ZnO target material with 50 µm (**a**) and 5 µm (**b**) scale bars.

**Figure 5 nanomaterials-13-00120-f005:**
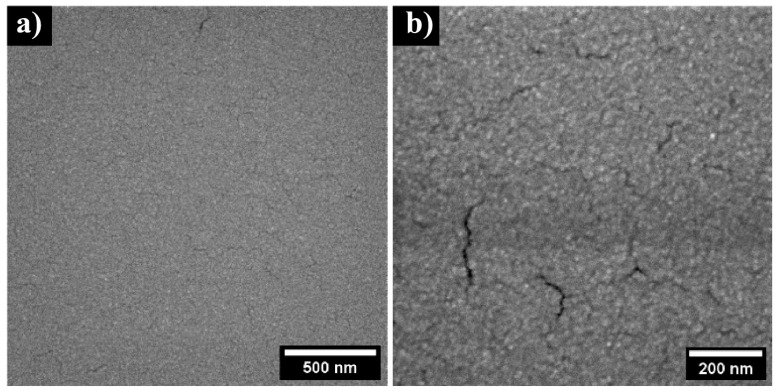
SEM images of the Fe_2_O_3_:ZnO film with 500 nm (**a**) and 200 nm (**b**) scale bars.

**Figure 6 nanomaterials-13-00120-f006:**
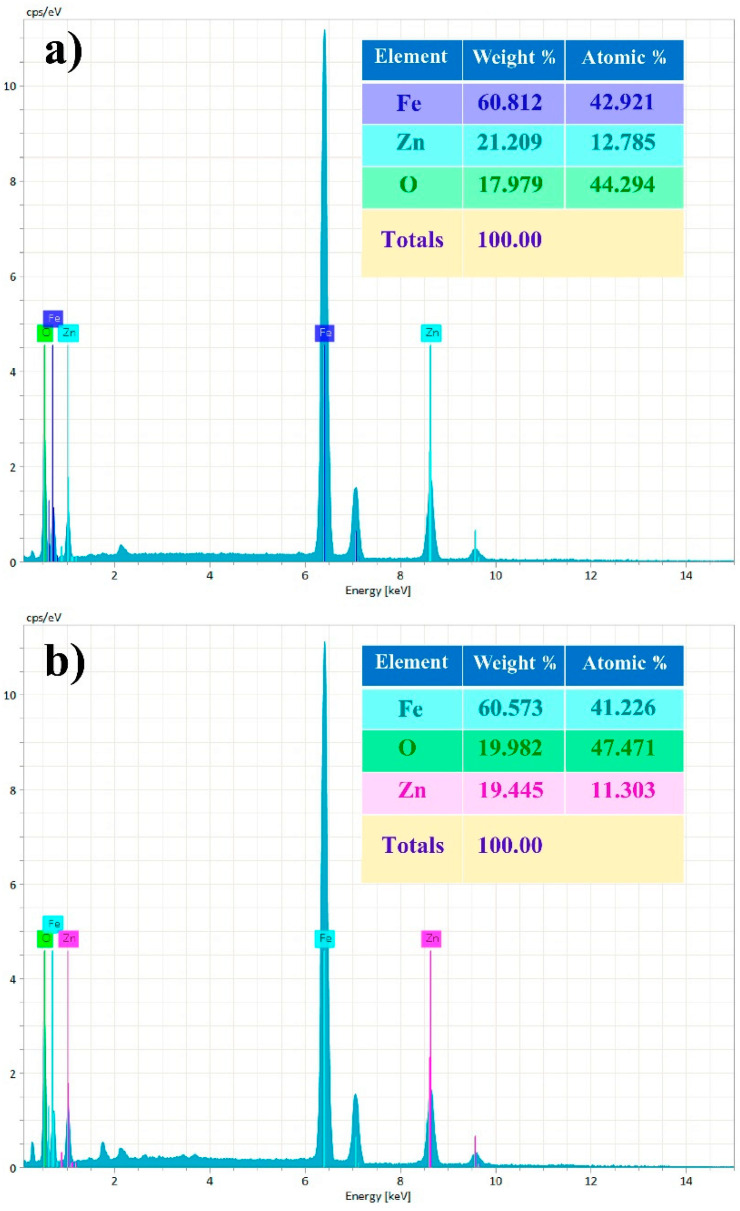
EDX spectrum of the Fe_2_O_3_:ZnO target (**a**) and the film (**b**).

**Figure 7 nanomaterials-13-00120-f007:**
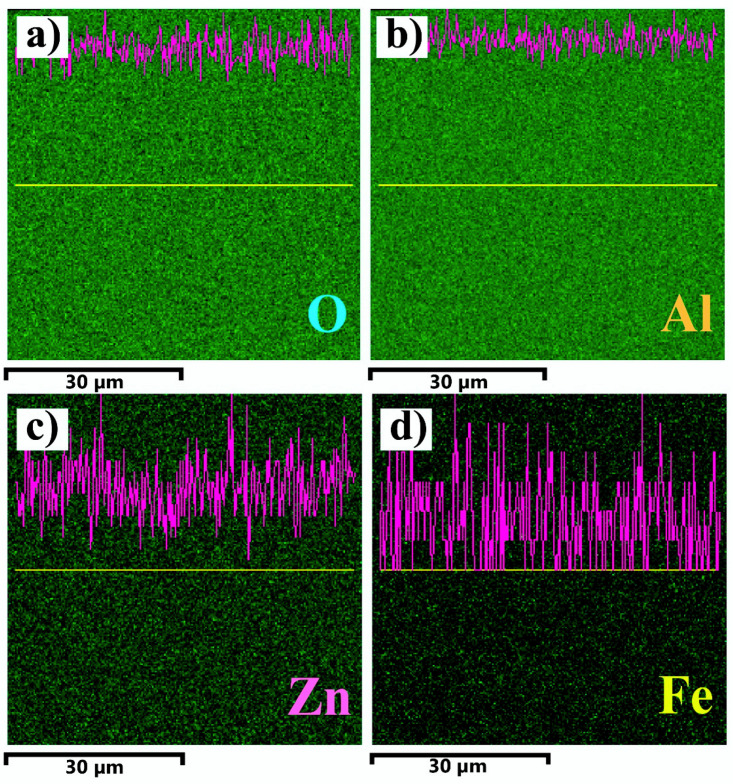
EDX elemental mapping analysis and X-ray intensity of characteristic lines of O (**a**), Al (**b**), Zn (**c**), and Fe (**d**) elements.

**Figure 8 nanomaterials-13-00120-f008:**
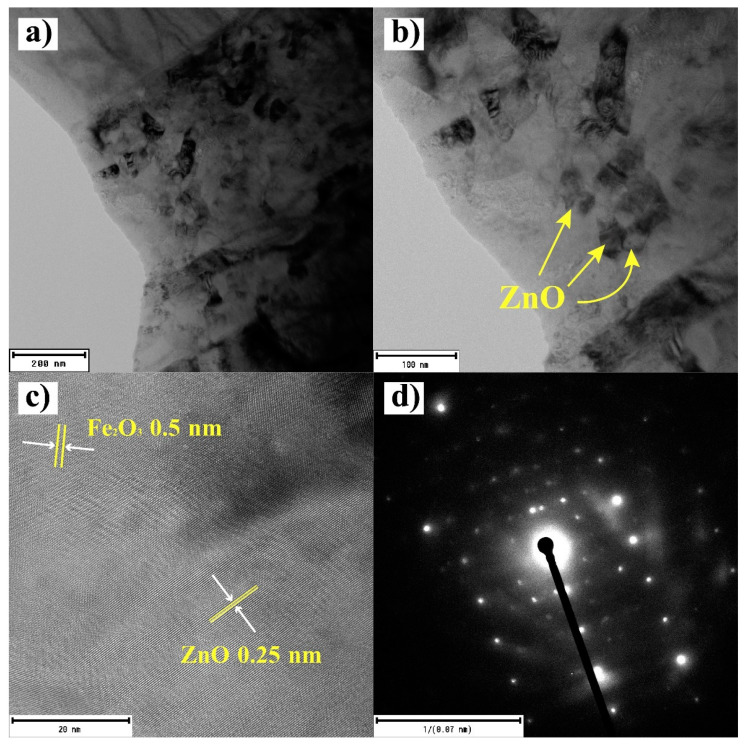
Low (**a**,**b**) and high (**c**) resolution TEM images and SAED pattern (**d**) for the Fe_2_O_3_:ZnO material.

**Figure 9 nanomaterials-13-00120-f009:**
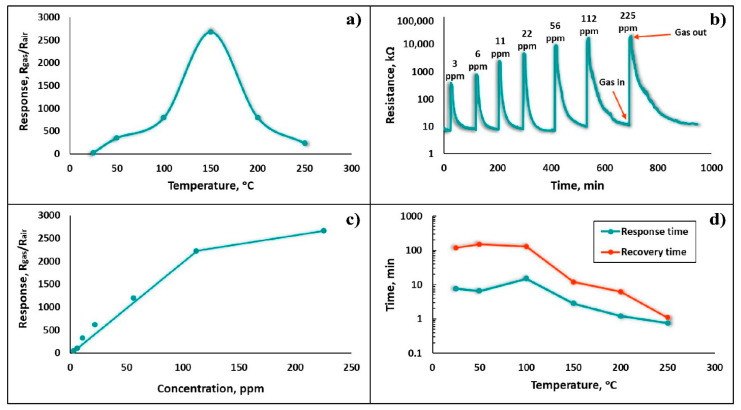
(**a**) Sensor response vs operating temperature at 225 ppm HPV, (**b**) dynamic resistance curves of the sensor to the different concentrations of HPV at 150 °C, (**c**) response vs HPV concentration, (**d**) dependence of the response and recovery times on the operating temperature.

**Figure 10 nanomaterials-13-00120-f010:**
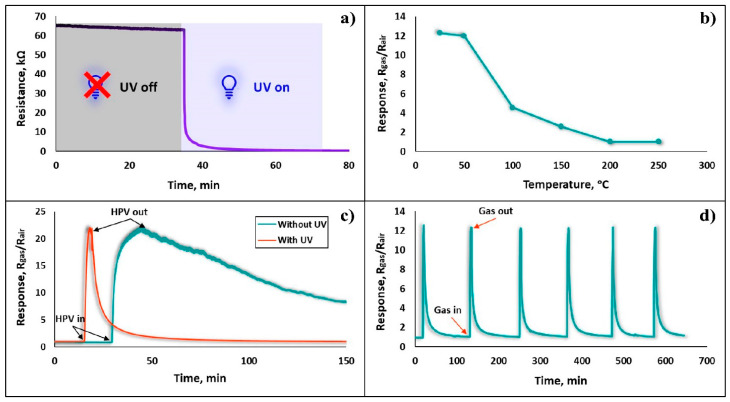
(**a**) UV response of the Fe_2_O_3_:ZnO sensor at RT, (**b**) sensor response vs operating temperature under UV irradiation, (**c**) dynamic response curves of the sensor to 3 and 225 ppm concentrations, with and without UV irradiation, respectively, (**d**) repeatability test of the sensor exposed to 1.5 ppm HPV under UV irradiation.

**Figure 11 nanomaterials-13-00120-f011:**
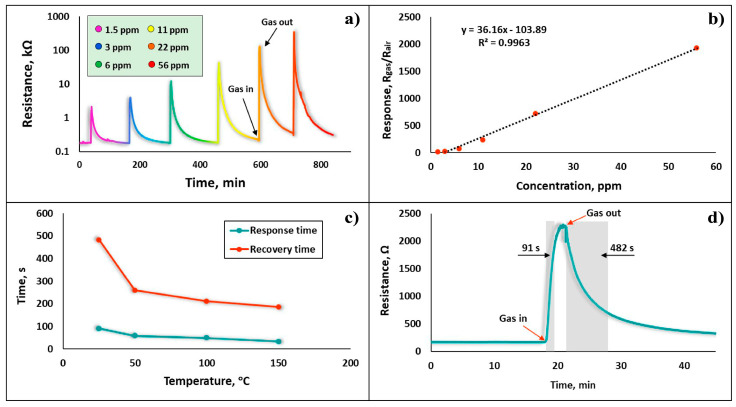
(**a**) dynamic resistance curves of the sensor to the different concentrations of HPV with UV irradiation, (**b**) sensor response vs HPV concentration under UV light, (**c**) dependence of the response and recovery times on the operating temperature under UV light, (**d**) real-time resistance curve of the sensor at 1.5 ppm HPV, representing the response and recovery time under UV irradiation at RT.

**Figure 12 nanomaterials-13-00120-f012:**
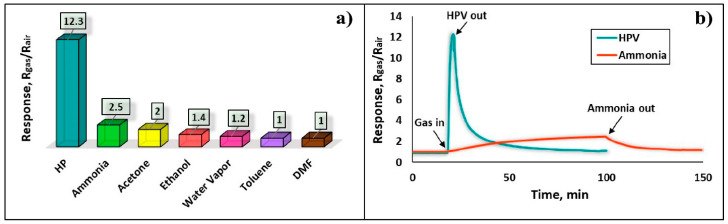
(**a**) Selectivity of the Fe_2_O_3_:ZnO sensor for HPV under UV irradiation at RT and (**b**) real-time response curves of the sensor to 1.5 ppm HPV and 130 ppm ammonia.

**Figure 13 nanomaterials-13-00120-f013:**
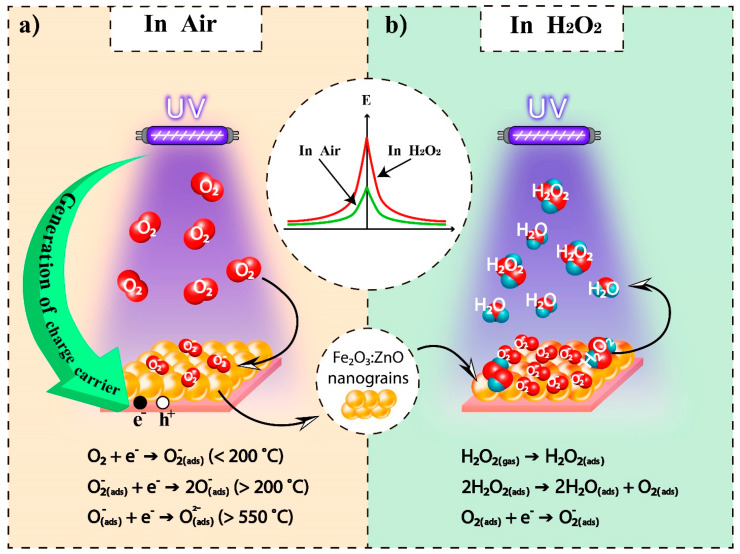
The schematic diagram of the proposed HPV sensing mechanism for the Fe_2_O_3_:ZnO sensor under UV irradiation in the air (**a**) and in HPV environment (**b**).

**Table 1 nanomaterials-13-00120-t001:** The deposition regimes of the Fe_2_O_3_:ZnO based thin layer and the Pd catalytic particles.

Process	Sputtering Duration	Power of Generator	Working Pressure	Sputtering Gas	Substrate Temperature	Cathode Current	Base Pressure
Magnetron sputtering (RF) (Fe_2_O_3_:ZnO layer)	25 min	70 Wt	2 × 10^−1^ Pa	Ar	200 °C	–	1 × 10^−3^ Pa
Magnetron sputtering (DC) (Pd catalytic particles)	5 s	–	5 × 10^−1^ Pa	Ar	200 °C	250 mA	3 × 10^−3^ Pa

**Table 2 nanomaterials-13-00120-t002:** The comparison of HPV sensing performance of the Fe_2_O_3_:ZnO material with the performance of previously reported sensors.

Materials	T (°C)	HPV (ppm)	Response	Reference
MnO_2_/Polyimide	140	20	30%	[34]
Porphyrin nanofiber/single-walled carbon nanotubes (SWCNTs)	RT	0.1	11.25%	[35]
(Pt-SWCNTs)	RT	2.6	2.7%	[36]
Porphyrin/polydimethylsiloxane (PDMS)/paper	RT	2.6	45.4	[37]
Tetrabutylammonium hydroxide (TBAH)	RT	0.013	25 %	[38]
Polyvinyl alcohol (PVA)/NaNO_2_	RT	5	14%	[39]
MoS_2_/reduced graphene oxide (RGO)	RT	50	12 (~373.1%)	[40]
Silver/gold metallic nanoparticles	RT	100	50%	[41]
NiO-en-PPy (polypyrrole) nanocomposite	RT	225	1.3	[42]
Fe_2_O_3_:ZnO nanograins	150	3	42	This work
Fe_2_O_3_:ZnO nanograins	RT	1.5	12	This work

## Data Availability

The data presented in this study are available on request from the corresponding authors.

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
