# Peer review of "Room Temperature Detection of Hydrogen Peroxide Vapor by Fe2O3:ZnO Nanograins"

_nanomaterials, 2022, doi:10.3390/nano13010120_

Round 1

Reviewer 1 Report

Authors reported a Fe2O3:ZnO sputtering target and a nanograins based sensor for room temperature (RT) detection of hydrogen peroxide vapor. This work is interesting and well-organized. I wonder this paper can be published after the minor revision. 

a. Authors seldomly illustrated the meachanism of Fe2O3:ZnO in response to HPV.

b. Authors didnot significantly focus on performances of the sensor under room temperature test

Author Response

Dear Reviewer,

First of all, thank you very much for the MINOR REVISION and for the new scientifically weighty considerations and remarks. All new corrections and additions have been marked in bold in the text.

Reviewer 2 Report

The authors present and interesting paper about the detection of H2O2 vapors by an F2O3:ZnO composite material. I think that the paper can be published on Nanomaterials after some revisions to the text:

1 - Page 3, lines 107-108: the Pd nanoparticles have never be mentioned in the abstract and in the introduction. What is their function?

2 - Page 5, lines 128-130: "As and oxidizing gas, HPV ...". I think this sentence is misleading. Fe2O3 and ZnO cannot clearly be further oxidized, so where is the sensor response coming from. This should be further explained.

3 - page 11, lines 234-235: "It is assumed that the effect of the UV rays .." This also should be better explained. The UV radiation decreases the resistance of the material because of the excitation of electrons in the CB of either  Fe2O3 or ZnO the most probably form some kind of nano-junction (Type II ?) that should again considered in the discussion.

4 - Paragraph 3.2: The description of the sensing mechanism is not clear or well done. If I understood well the authors assume a dismutation of H2O2 with O2 production that is then reduced to O2-on the sensor surface. What happens then to O2-? Are the vacancies on the oxide surface playing some role? Why is the Fe2O3:ZnO composite working better than ZnO or Fe2O3?

I think that the discussion should be extended and the mechanism better explained.

Author Response

(The authors gave the same response as above.)

Reviewer 3 Report

The authors investigate a Fe2O3/ZnO-based nanograin sensor for the detection of hydrogen peroxide vapor in low concentrations. The topic is interesting and relevant. The manuscript is clearly structured and largely also clearly written. Yet before publication I would recommend a revision regarding the following points:

-  Figures 5a and b do not really convey any information. It would have been more interesting to see the surface in higher resolution, but also Fig. 5d is a bit "blurred"

- the value for Zn in line 167 (19.98%) does not correspond to the one in the table in Fig. 6b (19.44%). Also, it's advisable to use the same number of digits after the comma when comparing such values, taking into consideration the possible errors in the measurement. It doesn't make sense to give 5 digits after the comma if the error is in the first digit.

- on page 8, a better explanation on how SAED works would be nice and why Fig. 8d proves that the material should be polycristalline. Also, in the caption of Fig. 8 there's a typo (SEAD)

- what is (23) in line 197?

- the statement in lines 161/162 that in a long-term interval would be "no significant drift" does somehow not correspond to the measurements shown in Fig. 9 and 10a, where a baseline drift (or hysteresis effect?) is clearly visible. Please include a statement on this.

- what is the labelling "response" on the y-axis of some measurements? Does it have a unit? Why not show the resistance in Ohms/kOhms? It is a bit like comparing apples and oranges like that

- in lines 248/249 you say "sensor consistently showed a stable response with nearly identical response/recovery curves". Can you quantify this? In Fig. 10d it does not look "nearly identical".

- in line 263 you say the sensor response in Fig. 11b would be "indisputably linear". I wouldn't say that. A quadratic fit would probably be much better....please clarify.

Author Response

(The authors gave the same response as above.)

Reviewer 4 Report

Ref.comments to the paper titled as “Room temperature detection of hydrogen peroxide vapor by  Fe2O3:ZnO nanograins” written by the authors: Mikayel Aleksanyan, Artak Sayunts, Gevorg Shahkhatuni, Zarine Simonyan, Hayk Kasparyan and Dušan  Kopecký.

Currently, more and more scientific and engineering groups are paying attention to the creation of the sensor devices using new materials, as well as novel approaches for testing and verifying properties. Naturally, such sensors are useful for detecting gases, impurities, and various foreign agents that harm humans and nature. From this point of view the manuscript is actual and modern. Really the Fe2O3:ZnO structure is a good target for this aim.

For the first, it is remarked that the authors have made good literature search, analyzing 47 references. This indicates the knowledge of the problem, its useful application and finding ways to solve it. Moreover, I have seen the deep analysis of the papers written by the last 5 years. It is good!

As for my general local opinion: The paper is interesting and prepared with good illustrations.

Experimental part. Gas sensor fabrication section.

Would you please to tell at which firms the α-Fe2O3 and ZnO materials, as the first step agents have been purchased? It should be say that the ZnO materials have produced by the different firms with the different initial parameters. The procedure to make the Fe2O3:ZnO sputtering target (High Vacuum Magnetron Plasma System) is general and it is not contradicted with known technical approach. It is interesting the visualization data shown all technology steps in Figure 2: Schematic block diagram of the HPV sensor fabrication.

Experimental part. Gas sensing test section.

It is good! In the text body please tilt the Latin symbols, for example, the sensor yield and resistances values.

Experimental part. Characterization section.

SEM images of the Fe2O3:ZnO target material and SEM images of the Fe2O3:ZnO film can show for researchers the average pore sizes in the target and homogenously obtained system as film. EDX spectrum of the Fe2O3:ZnO target and the film is also very informative data.

Results and discussions section.

Study of the sensor response depended on the operating temperature; dynamic resistance behavior, sensor response via the change of the HPV concentration, etc. are useful.

About the UV treatment. Generally, it is so strong influence of the UV conditions on the density of the materials and their energetic levels. Please explain in the details, how the refractive index can be change in your testing materials after UV use? May be you have created the grating in the materials body, that can influence on the sensor sensitive properties?

Data shown in Figure 12:Selectivity of the Fe2O3:ZnO sensor for HPV under UV irradiation at RT and real time response curves of the sensor to 1.5 ppm HPV and 130 ppm ammonia – can be used in the education process as well!

Conclusion part should be extended. It is not accumulate all good results obtained.

So, the paper is interesting for the specific area for the researchers and students. I can recommend to the authors to answer the questions mentioned above.  Thus, the paper can be published after minor corrections.

Author Response

(The authors gave the same response as above.)

Round 2

Reviewer 2 Report

The authors have answered all the points raised in my previous review and the paper is now clearer and ready to be published on Nanomaterials.

Reviewer 3 Report

Thanks to the authors for their revision of the manuscript weich can in my opinion be published as it is now.